# The role of refugee and migrant migration status on medication adherence: Mediation through illness perceptions

**Wejdan Shahin** ⊙ *, **Gerard A. Kennedy, Wendell Cockshaw, Ieva Stupans**

School of Health & Biomedical Sciences, RMIT University, Bundoora, Victoria, Australia

* wejdan.shahin@rmit.edu.au

**Data Availability Statement:** Data are available from figshare: https://doi.org/10.6084/m9.figshare. 11350085.

## Abstract

### Background

Illness perceptions may vary between different populations. This raises the question as to whether refugees and migrants of the same ethnic background have different perceptions. Understanding differences may have a significant impact on enhancing medication adherence in these groups.

### Objective

The study examined the associations and differences between illness perceptions, and medication adherence in hypertensive Middle Eastern migrants and refugees.

### Methods

Middle Eastern refugees and migrants ($\geq$30 years old), with hypertension were recruited from Arabic community groups in Australia and asked to complete a cross-sectional survey. The survey consisted of basic socio-demographic and clinical profile, self-reported illness perceptions, and self-reported medication adherence. The outcome measure was the Medication Adherence Questionnaire. Simple mediation modelling was applied to examine the role of illness perceptions as a mediator between different migration statuses, and medication adherence.

### Results

A total of 320 participants were recruited; 168 refugees, and 152 migrants. Educational level was found to be positively significantly associated with medication adherence in refugees, $p = 0.003$, while employed migrants were more likely to report higher adherence to hypertensive medication, $p = 0.005$. In both groups, there was a significant association between illness perceptions and medication adherence $p = 0.0001$. Significant differences were found between both groups regarding adherence and illness perceptions variables. Refugees had more negative illness perceptions and were less adherent than migrants. Illness perception was a mediator in the relationship between migration status and medication adherence; the

                                    

**Funding:** The authors received no specific funding for this work.

**Competing interests:** The authors have declared that no competing interests exist.

unstandardized indirect effect was 0.24, and the 95% confidence interval ranged from (0.21–0.36).

## Conclusions

To achieve better adherence to medications in vulnerable populations such as refugees, illness perceptions need to be understood, and differentiated from other populations, such as migrants from similar backgrounds. Patients' education about illnesses and medications should be specific and targeted to each population. Interventional studies are recommended to modify refugees' and migrants' illness perceptions, to enhance medication adherence and wellbeing.

## 1. Introduction

Hypertension is one of the most common preventable causes of death worldwide. It is estimated that about a quarter of adults in the world have hypertension [1]. The effective control of hypertension requires patients to take medication regularly and to maintain a healthy lifestyle [2]. However, effective control presents particular challenges when trying to maintain long-term patient adherence and achieve therapeutic goals, due to the often-asymptomatic nature of hypertension [3]. Unfortunately, medication adherence rates are generally only between 50%-70% [2].

Medication adherence is defined as the process by which patients take their medication as prescribed [4]. However, some patients may not understand the directions correctly, and/or may decide not to take medication as recommended [5]. Many factors may be positively associated with medication adherence, such as education, employment, and age. Ethnic minorities, higher medication costs, and regimen complexity may have negative effect on medication adherence [6]. Medication adherence goes beyond medication consumption and is a reflection of healthy behaviour [7]. Thus, patients' acceptance of medical advice, including medication use, may be influenced by subjective beliefs about diseases [8]. A number of studies have also assumed the view that disease may be a response to social stresses and/or life events and is shaped in part by the nature of the cultural label which is applied to a person's condition [9].

Theoretical models of patient behaviour can be useful in designing interventions to improve medication adherence [10]. In the Common-Sense Model of Illness Perception, patients make sense of their symptoms by forming causal attributions about the illness, how long they think the illness will last, whether it can be controlled or cured, and what consequences the symptoms of the illness will have [3]. According to Leventhal and his colleagues [11], illness perception consists of five factors: (1) 'identity' (label of illness and symptoms); (2) 'timeline' (duration of illness including symptoms and recovery); (3) 'consequences' (the seriousness of the disease); (4) 'control' (amenability of the illness to being cured, prevented or treated); and (5) 'causes' (possible causes of the illness). A sixth factor, illness coherence, has been added more recently to this model to represent overall patient understanding of the illness [12]. Two relatively recent systematic reviews [8, 10] have described a significant association between illness perception domains and medication adherence but have also revealed an inconsistency in the direction (positive or negative) of the associations in different studies. Thus, further work is required to clarify the direction of the relationships between illness perception domains and medication adherence.

Australia has been involved in the UNHCR (United Nations High Commission for Refugees) resettlement program since 1977 and has consistently ranked as one of the top three resettlement countries in the world [13]. Over the past 50 years, conflicts in Lebanon, Algeria, Sudan, Libya, Iraq, Syria and Kuwait have collectively resulted in many hundreds of thousands of refugees seeking safety in neighbouring states and in more distant countries. The Arab Spring uprisings have now contributed millions more refugees and migrants to this exodus [14].

There is increasing evidence that immigrants and traumatized refugees have an elevated prevalence of medical diseases, such as hypertension [15]. Increased vulnerability to physical, mental and social health problems may result from the process and the specific circumstances of migration [16]. Thus, the increasing numbers of refugees and migrants arriving from the Middle East underscores the need to better understand chronic illnesses and medication adherence in this population.

In the literature refugees and migrants have been considered as single population and have been treated under the same umbrella. However, each represents different populations that can be distinguished from each other. A refugee is defined by the 1951 Refugee Convention as "a person who, 'owing to a well-founded fear of persecution for reasons of race, religion, nationality, membership of a particular social group or political opinions, is outside the country of his/her nationality and is unable or, owing to such fear, is unwilling to avail themselves of the protection of that country." On the other hand the International Organization for Migration (IOM) defines a migrant as "any person who is moving or has moved across an international border or within a State away from his/her habitual place of residence, regardless of (a) the person's legal status, (b) whether the movement is voluntary or involuntary, (c) what the causes for the movement are or (d) what the length of the stay is" ([17]p.2). Although, the both groups may have similar difficulties during the resettlement process, there are distinct differences between those who migrate voluntarily and those who have little or no choice in the matter. The intentions and motivations for migration differ vastly between refugees and migrants [18]. The two groups can be distinguished by the fact that refugees cannot safely return home, because of a threat of persecution or death, but migrants face no such impediment to returning to their country of origin [19]. Furthermore, refugees must leave their home countries quickly, meaning that they are largely unprepared for the journey ahead and this may further exacerbate feelings of having little or no control over their lives.

In contrast, migrants may feel like they are gaining control over their lives through migration [18]. In addition, refugees have been forced into a situation where responsibility for and control over their own lives has been taken away from them. Their existence and future are uncertain, and many experience a constant fear of being deported. The powerlessness they experienced generates uncertainty that has negative implications for health [20]. Unemployment, and denial of access to health services are risk factors for psychiatric morbidity, and chronicity of health conditions. Refugees constitute a particularly high risk group [21]. These factors differentiate the groups and there are likely to be significant differences between migrants and refugees across a number of their personal and social issues, taking into account the damaging effect of the war on the education, employment, socioeconomic status and refugees' health generally [22]. Therefore, these two different populations might develop different health beliefs, and perceptions about the same illness. Thus, it is important to have a well-founded insight into how Middle Eastern refugees and migrants perceive chronic conditions such as hypertension, and to understand how their perceptions may influence medication adherence behaviours.

A review of studies which addressed medication adherence in the Middle Eastern population with chronic illnesses, such as hypertension, diabetes and chronic obstructive pulmonary

disease, demonstrated that participants' self-reports gave an estimate of 48% of non-adherence [23]. A study of 392 Middle Eastern Arabic-speaking migrants and refugees in Australia investigated diabetes self-care activities, and found that 88% of Arabic- speaking participants were non-adherent to prescribed medication, in comparison with 45.1% of 309 English speaking Caucasian Australian participants [24]. Aside from these studies, only several others have examined medication adherence and/or illness perceptions in a Middle Eastern population [3, 25]. To the best of the authors' knowledge, there have been no previous studies on this topic in hypertensive patients, conducted in Australia or indeed anywhere else in the world. Similarly, there have been no previous studies which have assessed the differences between refugees and migrants regarding illness perceptions and the impact residency status has on medication adherence.

The objectives of this study were to examine migration status (refugee or migrant) differences in medication adherence, and to test the mediating role of illness perceptions in Middle Eastern refugees and migrants in Australia with hypertension.

## 2. Materials and methods

### 2.1 Study design and setting

A cross-sectional design was used in this study. After obtaining approval from RMIT University Ethics Committee, (SEHAPP 53–18) data were collected from September 2018 to July 2019 using a convenience sampling process.

Participants were recruited from various community groups established by non-profit Australian organisations in Melbourne, where Middle Eastern refugees and migrants meet to share their interests and gain support from members of their community. They were also recruited through an Adult Migrant English Program, where new migrants and refugees learn foundation English to enable them to participate socially and economically in Australian society. Arabic community groups in various states of Australia, available on Face-book, were also used to recruit refugees and migrants. Participants were given access to the participant information sheet which explained the study. The completion of the anonymous survey implied consent.

### 2.2 Study participants

A total of 319 participants were recruited; 168 refugees, and 151 migrants. Participants' demographic characteristics are described in Table 1.

Throughout the 10-month recruitment period, attendees at the community groups, or Adult Migrant English Program centres were approached and invited to consider participating in the study. A poster including the survey link was published in some Facebook Arabic interest gathering groups in Australia. Those who met the following criteria were invited to participate in this study: (1) aged 30 years or older; (2) migrated to Australia as refugee or migrant; (3) born in any of the 22 countries of the Middle East; and (4) diagnosed with essential hypertension. Individuals who were younger than 30 years old, unable to speak English, or Arabic and originally not from Middle East were excluded. Regarding migration status, participants were asked to nominate one of the following responses described how they arrived in Australia: "*refugee*," "*work*," "*studying*," "*economic reasons*" "*any other reason*". Participants who selected an answer other than "refugee" were considered migrants.

### 2.3 Development of questionnaire

The self-report questionnaire consisted of 19 items in 4 sections. The first section was comprised of socio-demographic information including age, gender, place of birth,

**Table 1. Demographics and clinical characteristics for refugees and migrants (n = 319).**

| Variables | | Refugee *n* = 168 | Migrant *n* = 151 | $\chi$ (*df*) | *p* |
|---|---|---|---|---|---|
| | | *n* (%) | *n* (%) | | |
| Age | 30–40 | 23 (13.8%) | 29 (19.2%) | 20.78(3) | 0.001 |
| | 41–50 | 35 (21%) | 59 (39.1%) | | |
| | Above 50 | 108 (64.7%) | 60 (39.7%) | | |
| | Missing | 1 (0.6%) | 1 (0.66%) | | |
| Sex | Male | 83 (49.4%) | 64 (42.4%) | 1.58(1) | 0.20 |
| | Female | 85 (50.6%) | 87 (57.6%) | | |
| Education | Lower secondary | 88 (53.7%) | 42 (28.4%) | 40.57(4) | 0.0001 |
| | Higher secondary | 41 (25%) | 26 (17.6%) | | |
| | Diploma | 7 (4.3%) | 18 (12.2%) | | |
| | Bachelor | 22 (13.4%) | 34 (23%) | | |
| | Higher than bachelor | 6 (3.7%) | 28 (18.9%) | | |
| | Missing | 4 (2.3%) | 3 (1.98%) | | |
| Occupation | Home/Not working | 139 (84.8%) | 84 (55.6%) | 38.35(2) | 0.001 |
| | Self-employer | 4 (2.4%) | 31 (20.5%) | | |
| | Governmental/private | 21 (12.8%) | 36 (23.8%) | | |
| | Missing | 4 (2.3%) | - | | |
| Arrival year to Australia | 2015–2018 | 58 (34.7%) | 23 (15.4%) | 24.35(3) | 0.0001 |
| | 2010–2015 | 55 (32.9%) | 42 (28.2%) | | |
| | 2000–2010 | 33 (19.8%) | 41 (27.5%) | | |
| | Before 2000 | 21 (12.6%) | 43 (28.9%) | | |
| | Missing | 1 (0.6%) | 2 (1.3%) | | |
| Co-morbidities | Having ≥ 2 chronic illnesses | 54 (32.1%) | 35 (23.2%) | 5.5 (1) | 0.02 |
| | Diabetes Mellitus | 61 (39.4%) | 38 (25.7%) | 6.44 (1) | 0.01 |
| | Mental illness | 12 (7.4%) | 3 (2%) | 4.98 (1) | 0.03 |
| | COPD | 7 (4.2%) | 6 (4%) | 0.01 (1) | 0.9 |
| | Asthma | 16 (10.3%) | 14 (9.5%) | 0.06 (1) | 0.8 |
| | Back pain | 57 (35.4%) | 42 (28%) | 1.96 (1) | 0.16 |
| | Arthritis | 42 (26.3%) | 36 (24.2%) | 0.18 (1) | 0.67 |
| Country of birth | Iraq | 83 (49.4%) | 17 (11.2%) | - | - |
| | Syria | 54 (32.1%) | 18 (11.8%) | - | - |
| | Lebanon | 17 (10.12%) | 45 (29.6%) | - | - |
| | Egypt | 3 (1.8%) | 18 (11.8%) | - | - |
| | Morocco | 2 (1.2%) | 11 (7.23%) | - | - |
| | Jordan | NA | 13 (8.55%) | - | - |
| | Algeria | 1 (0.6%) | 5 (3.3%) | - | - |
| | Kuwait | NA | 9 (6.3%) | - | - |
| | Emirates | NA | 4 (2.8%) | - | - |
| | Saudi Arabia | NA | 4 (2.8%) | - | - |
| | Other Arab countries | 6 (3.6%) | 8 (5.3%) | - | - |

education level, and occupation. The second section canvassed the major chronic conditions identified by the Australian Institute of Health and Welfare: arthritis, asthma, back pain/problems, cancer, cardiovascular disease (such as coronary heart disease and stroke), chronic obstructive pulmonary disease (COPD), diabetes and mental health conditions [26].

The last two sections, used validated, and reliable tools to measure medication adherence [27] and illness perceptions [28]. Content validity of the questionnaire was examined by review by three academic researchers.

The questionnaire was available in English and Arabic. The questionnaire was translated into Arabic by a researcher whose first language was Arabic, then back-translated to English by another bilingual researcher. The original questionnaires were compared with the back-translated version by two researchers whose first language was English, and no discernible differences were detected.

## 2.4 Sample size

The sample size was calculated using Gpower* software version three.

With alpha set at $p < 0.05$, (two-tailed) and power at 0.95, the estimated minimum sample size was calculated to be 105 participants. A total of 319 participants were recruited which exceeded the number participants required to detect significant differences and relationships by a factor of three.

The effect size was measured using Cohen's *d* statistic. Consistent with the literature, we used established cut-offs of 0.2, 0.5, and 0.8 for a small, moderate, and large effect sizes respectively [28].

## 2.5.1 Medication Adherence Questionnaire

Medication adherence was measured using the Medication Adherence Questionnaire (MAQ), a questionnaire adapted from the Morisky self-reported medication adherence questionnaire relating to medication use and major reasons for non-adherence. The four-item MAQ was selected because it has been well-validated to identify adherence behaviour in a number of chronic cardiovascular disease populations and scores have been shown to correlate well with objective adherence measures and clinical outcomes, such as blood pressure, lipid levels and blood glucose control [29]. The psychometric properties of this questionnaire ranged from adequate [27, 30] to high [31] in different studies. The MAQ measures both intentional and unintentional non-adherence based on forgetfulness, carelessness, stopping medication when feeling better and stopping medication when feeling worse. The scale is scored 1 point for each "no" and 0 points for each "yes". Patients were described as adherent (if the total score was four) or non-adherent (if the total score was less than 4) [32, 33].

## 2.5.2 Brief illness perceptions questionnaire

Hypertension Illness perceptions were assessed using the Brief Illness Perceptions Questionnaire (BIPQ). This questionnaire was selected as it provides simple and rapid assessment of illness perceptions. The BIPQ has advantages in terms of brevity and lower participant burden. It also demonstrated good psychometric properties, including concurrent, predictive and discriminant validity, and it has been widely used with different chronic conditions [34]. It contains seven items to assess perceptions according to the domains of Common Sense Model [28]. Each item of "personal control", "treatment control", and "coherence," "identity," "timeline" and "consequences" was scored on a scale of 1–5; "1" for strongly disagree, and "5" for strongly agree. Scoring for "consequences" and "identity" were reversed. High scores indicate positive perceptions of hypertension, whereas lower scores show negative illness perceptions, with negative impacts on life, and experiences of severe symptoms of illness respectively. Causality of hypertension was a free text section. Participants were asked to list the three most likely causes for their hypertension. Answers were scored by three researcher WS, GK, and IS on a scale of 0–2. Evidence from the literature was used to compare the relevance of the causes

specified [35]. A score of two points was given to causes similar to those reported in the literature, for example, obesity, heredity, or stress. For causes that may be related to stress, such as fear from war, not finding work, or economic status, a score of one was given. A score of zero was given for irrelevant causes, such as fate, weather (Table 2). To assess inter-rater reliability, Intra Class Correlation (ICC) estimates and their 95% confident intervals were calculated based on absolute-agreement, and 2-way mixed-effects model (ICC = 0.93, CI 0.84–0.97, p = 0.0001).

## 2.6 Data analysis

Data were analysed using the IBM Statistical Package for the Social Sciences software (Ver. 26) for Windows. The internal reliability of MAQ was assessed using Kuder-Richardson's coefficient (KR20), which measures internal consistency of questionnaires feature dichotomous items [36] (0.76), and the BIPQ was assessed using Cronbach's alpha (0.79). Descriptive statistics including frequencies, percentages, means and standard deviations examined participants' socio-demographics characteristics, and all variables. Comparisons of dependent variables between the two groups were made using Chi-square tests or independent-samples *t*-tests as appropriate (Levene's test was used to assess homogeneity of variance). Bivariate associations between dependent variables were examined using Pearson's correlations (*r*). A two-tailed significance level of 0.05 was used for statistical procedures.

**Table 2. Refugee and Migrant causal attributions for hypertension.**

| Status | Rank | Causes | Score | %(*n*) |
|---|---|---|---|---|
| **Refugee** | 1 | Stress | 2 | (31) 31% |
| | 2 | Fear from war | 1 | (18) 18% |
| | 3 | Fate | 0 | (11) 11% |
| | 4 | Don't know | 0 | (8) 8% |
| | | Heredity | 2 | (8) 8% |
| | 5 | Close relatives death | 1 | (7) 7% |
| | 6 | Depression | 1 | (5) 5% |
| | 7 | Weather | 0 | (3) 3% |
| | 8 | Not speaking English | 1 | (2) 2% |
| | | Salty food | 2 | (2) 2% |
| | | Migration | 1 | (2) 2% |
| | 9 | Not finding work | 1 | (1) 1% |
| | | physical inactivity | 2 | (1) 1% |
| | | Smoking | 2 | (1) % |
| **Migrant** | 1 | Stress | 2 | (33) 36.7% |
| | 2 | Heredity | 2 | (23) 25.6% |
| | 3 | Obesity | 2 | (10) 11.1% |
| | 4 | Salty food | 2 | (6) 6.7% |
| | 5 | Don't know | 0 | (5) 5.6% |
| | 6 | Having DM | 2 | (4) 4.4% |
| | 7 | Physical inactivity | 2 | (2) 2.2% |
| | | Smoking | 2 | (2) 2.2% |
| | | Aging | 2 | (2) 2.2% |
| | | Economic reasons | 1 | (2) 2.2% |

Relevant = 2, partially relevant = 1, not relevant = 0.

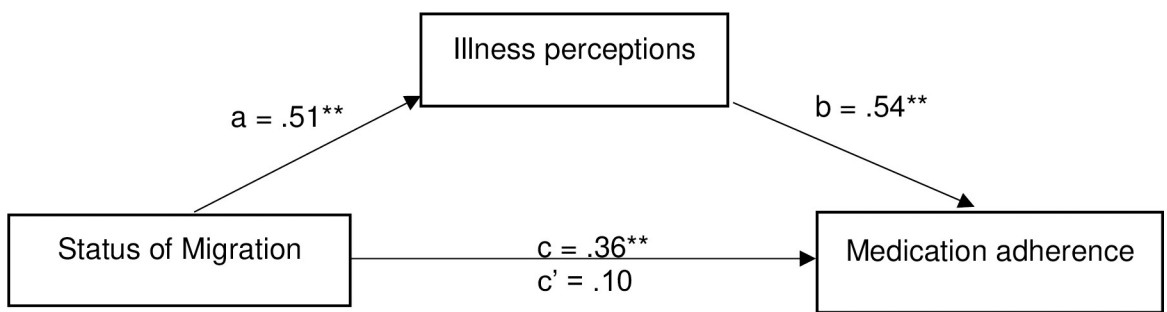

**Fig 1. Mediation effects of illness perceptions on the relationship between statuses of migration and standardised path weights presented.**

A mediation model in which illness perception mediates the association between migration status (refugee or migrant) and medication adherence as presented in Fig 1 was tested. We applied bootstrapping (5,000 samples) using the SPSS PROCESS macro [36, 37], to analyse the model and determine the confidence interval for the indirect effect. This procedure does not require the indirect effect to be normally distributed, therefore is preferred to the Sobel's test [38] [39]. The indirect effect is statistically significant if the 95% bias-corrected confidence interval does not include zero.

Possible confounding factors that were significantly correlated to medication adherence were entered as covariates in the mediation analysis.

The dimensionality of the BIPQ items in the present sample was examined using factor analysis. The Kaiser-Meyer-Olkin measure of sampling adequacy was .76, above the commonly recommended value of .6, and Bartlett's test of sphericity was significant ($\chi^2 = 364$, $p < .0001$). The communalities were all above .6. Given these overall indicators, factor analysis was deemed appropriate.

## 3. Results

### 3.1 Participants demographics and clinical characteristics

A total of 320 participants were recruited; 168 refugees, and 152 migrants. Participants' demographic characteristics are described in Table 1. All participants were born in the Middle East, and there were slightly more women than men in both groups. The highest proportion of refugees were from Iraq and Syria. Overall, hypertensive migrants were significantly younger ($\chi^2 = 20.78$, $p = 0.001$), more likely to be employed ($\chi^2 = 38.35$, $p = 0.0001$), and were also significantly more likely to have a higher level of education ($\chi^2 = 40.57$, $p = 0.0001$) than hypertensive refugees. Migrants reported a significantly lower level of co-morbid conditions that are commonly associated with hypertension, e.g., diabetes mellitus ($\chi^2 = 6.44$, $p = 0.01$), and were significantly less likely to have mental health issues ($\chi^2 = 4.98$, $p = 0.03$).

### 3.2 Participants perceptions about hypertension and medication adherence

Across all items of the BIPQ, significant differences were found between refugees and migrants (Table 3). Refugees reported stronger negative beliefs about hypertension in comparison with migrants. For example, refugees showed a significantly higher identity perception ($p = 0.0001$) indicating that they attributed a lot of the symptoms they experienced to their hypertension. They also demonstrated significantly higher consequences perceptions ($p = 0.0001$), believing that their hypertension had a considerable negative influence on their lives. Migrants were significantly more likely to have higher positive beliefs about hypertension than refugees. For

**Table 3. Comparisons of refugee and migrant illness perceptions, and medication adherence.**

| BIPQ | Refugee M(SD) | Migrant M(SD) | t(df) | p |
|---|---|---|---|---|
| Illness perceptions (one factor) | 12.8(3.9) | 17.9(4.4) | 10.9 (298) | 0.0001 |
| Personal control | 2.88 (0.98) | 3.64(1.09) | 6.47 (302) | 0.0001 |
| Treatment control | 2.89 (1.23) | 3.91 (1.26) | 7.25 (306.5) | 0.0001 |
| Coherence | 2.78 (1.13) | 3.82 (1.18) | 7.86 (296.8) | 0.0001 |
| Illness identity | 3.82 (1.16) | 2.65 (1.26) | 8.56 (302) | 0.0001 |
| Illness consequences | 3.88 (1.24) | 2.8 (1.23) | 7.72 (311) | 0.0001 |
| Causes | 1.08 (0.79) | 1.73 (0.57) | 7.13 (224) | 0.0001 |
| Timeline | 2.8 (0.86) | 3.08 (0.53) | 3.22 (298) | 0.0001 |
| Medication adherence | 1.36 (1.4) | 2.5 (1.4) | 7.26 (305) | 0.0001 |

instance, they showed better coherence (understanding of hypertension, $p = 0.0001$), reporting a higher perception of their personal ability to control hypertension ($p = 0.0001$), and treatment control of their illness ($p = 0.0001$). Regarding the causes domain, a significant difference was found between both groups regarding their beliefs about the causes of hypertension ($p = 0.0001$) (Table 3). The three most commonly indicated causes for hypertension in the refugee group, were stress (31%), fearing war (18%), and fate (11%). Migrants attributed their hypertension to having stress (36.7%), heredity (25.6%) and obesity (11.1%) (Table 2).

Significant difference between refugees and migrant was found regarding medication adherence $p = 0.0001$. Migrants were more likely adherent to taking medications in comparison to refugees (Table 3).

## 3.3 Correlation of demographics/illness perceptions with medication adherence

Table 4 shows the results of correlational analyses for both groups. There was a weak, but significant correlation between higher education level and better medication adherence in the refugee group, $p = 0.002$, while employed migrants were more likely to report higher adherence to hypertensive medication, $p = 0.005$. Illness perception domains and medication adherence were significantly correlated in migrants and refugees. Positive perceptions about hypertension were associated significantly and positively with medication adherence in both groups, $p = 0.0001$. Participants who had higher control over their illness, and positive perceptions about treatment control, reported better medication adherence in both the refugee and migrant groups $p = 0.0001$ for both items across both groups. In addition, better understanding of hypertension was positively associated with medication adherence in refugees $p = 0.0001$ for both items across both groups. Negative perceptions about hypertension, such as consequences and identity were found to have negative influence on medication adherence in refugees $p = 0.007$, $p = 0.001$ and migrants $p = 0.0001$, $p = 0.001$ respectively. Those who attributed hypertension to its actual causes, also had a better understanding of their hypertension (coherence) $p = 0.0001$ and this was positively correlated with medication adherence in both refugees $p = 0.0001$, and migrants $p = 0.0001$. Table 4 shows the statistical details.

## 3.4 Illness perceptions as a mediator between migration status and illness perceptions

The relationship between different migration status and medication adherence was mediated by illness perceptions, after adjusting educational level, and employment. The results showed, that the standardized regression coefficient between migration status and illness perceptions

**Table 4. Correlations between medication adherence scores and other variables in refugees and migrant.**

| Variables | MAQ refugee | | MAQ migrants | |
|---|---|---|---|---|
| | r | p | r | p |
| Age | 0.06 | 0.43 | -0.009 | 0.9 |
| Gender | 0.1 | 0.2 | -0.03 | 0.77 |
| Employment | 0.14 | 0.07 | 0.23 | **0.005** |
| Education | 0.24 | **0.002** | 0.036 | 0.67 |
| Arrival year | -0.93 | 0.24 | -0.11 | 0.17 |
| Comorbidity | -0.11 | 0.18 | 0.04 | 0.62 |
| Illness perceptions (one factor) | 0.48 | **0.0001** | 0.53 | **0.0001** |
| Personal control | 0.33 | **0.0001** | 0.51 | **0.0001** |
| Treatment control | 0.44 | **0.0001** | 0.41 | **0.0001** |
| Causes | 0.43 | **0.0001** | 0.45 | **0.0001** |
| Timeline | 0.13 | 0.112 | 0.07 | 0.43 |
| Consequences | -0.22 | **0.007** | -0.31 | **0.0001** |
| Identity | -0.27 | **0.001** | -0.30 | **0.001** |
| Coherence | 0.4 | **0.0001** | **0.33**[**] | **0.0001** |

was statistically significant $p = 0.0001$, as was the standardized regression coefficient between illness perceptions and medication adherence $p = 0.0001$. We tested the significance of this indirect effect using bootstrapping procedures. Unstandardized indirect effect was 0.24, and the 95% confidence interval ranged from (0.21–0.36). Thus, the indirect effect was statistically significant (Table 5, Fig 1).

## 4. Discussion

This study is the first to investigate the association of migration status on medication adherence in hypertension mediated by Middle Eastern refugees' and migrants' illness perceptions, in Australia. The findings of this study indicated that the illness perception cognitive domain had a significant impact on medication adherence in both groups; refugees and migrants. The findings were consistent with the theoretical prediction of the behavioural Common Sense Model of illness perceptions [40], and the findings of previous studies [25, 41]. The evaluation of these perceptions in lesser studied populations such as refugees and migrants from the Middle East, is fundamental to developing specific and targeted interventions to improve medication taking behaviours. The second significant result from this study, was the identification of differences between refugees and migrants regarding their perceptions of hypertension and adhering to taking medications. Middle Eastern refugees were significantly more likely to perceive illness negatively and reported lower medication adherence rates than migrants from the same regions.

The results of this study are aligned with a previous research article that demonstrated the differences between Middle Eastern diabetic patients, and Caucasian English speaking diabetic patients, with regards to their illness, treatment perceptions and self-management adherence [24]. Middle Eastern Arabic speaking participants had lower medication adherence and

**Table 5. Bootstrap analyses of the magnitude and statistical significance of indirect effect.**

| Independent variable | Dependent variable | Mediator variable | Unstandardized indirect effect | Size effect | 95% CI mean indirect effect (lower and upper) |
|---|---|---|---|---|---|
| Status of migration | Adherence | Illness perceptions | 0.24 | 0.04 | 0.21–0.36 |

negative illness perceptions in comparison to the Caucasian English Speaking participants, however it is important to note that in this study [24] migration status was not taken into account.

Previous studies have not clarified the relationship between medication adherence and perceived causes of hypertension [2, 42], and causes have been excluded in other studies that have investigated the role of illness perceptions in predicting adherence to medications [24, 43]. In our study, causal attribution was associated positively with medication adherence in both refugees and migrants. The findings indicate that those who tend to attribute their hypertension causes to external factors, such as fate, or the weather were less likely to adhere to their therapeutic regimen. This may be explained through Leventhal and colleagues' work [44], that indicates that when a patient labels any illness, they will search for causes to attribute their illness, and these causes correspondingly shape their actions to cope with it. For example, many refugees who attributed their hypertension to the cause of "fate" may not be motivated to seek medical advice from health professionals. They may use their prayers to God as a treatment modality rather than using medicines [24]. Whereas, most of the migrants believed their illness was caused by risk factors, and therefore they may assume the explanation of illness attribution from the healthcare providers' perspectives. In addition, patients who believe that their illness is caused by external factors may perceive there to be less controllability of behavioural outcomes, and thus be less likely to adhere to their medications [45].

In this study treatment control was a significant predictor of medication adherence in refugees. When patients believe treatment could improve their symptoms, they were more likely to adhere to the doctor's prescription [2]. Although, the illness that we focused on in our study was hypertension, which has an asymptomatic nature, and most patients are labelled as being hypertensive after blood pressure screening without experiencing any symptoms [1], our findings indicated that refugees perceived significant symptoms that they attributed to hypertension, in comparison to migrants. These symptoms were associated with lower level of medication adherence. This may explain why treatment control was the most significant variable associated with medication adherence among refugees who attributed their blood pressure to causes using their personal beliefs about environmental or supernatural factors to make sense of the ambiguous symptoms.

The findings described in this paper are consistent with research from literature which has examined illness perceptions, and has indicated that patients who believe in their ability to control illness, are more likely to seek treatment and engage in healthcare behaviours, and consequently adhere to taking medications [1, 41, 45]. Our findings identified that personal control was the most significant predictor of medication adherence for migrants. Also, the findings suggested that migrants acquire better illness understanding than refugees, and therefore they have more confidence in their ability to affect hypertension, through their own personal control. Throughout the literature the link between personal control and health is well established. Patients with high personal control are more likely to have a healthy lifestyle, and they are more likely to seek and follow medical advice when ill [46]. Thus, better medication adherence would be expected when people have higher levels of personal control. Furthermore, based on the experience of migration for refugees and migrants, the latter gained more control over their lives, including their health and illness [18], and they are more skilled at coping with life crises that occur [46]. More positive beliefs about the sense of internal control have been associated with coping strategies, which are generally considered more adaptive such as positive reinterpretation, seeking social support and actively trying to tackle the problem [47]. Traumatized refugees who experienced war, forced migration or violence perceive absence of control over their lives—this can contribute to poor health as diet, exercise and medical treatment are neglected [48].

Social support plays an important role in determining treatment uptake, recovery and adherence [49]. Refugees who have been taken away from their friends and families, lack social support thus, worse adherence to medication and health recovery would be expected. In literature, reduced posttraumatic stress was associated with securing work rights and health cover. Living in the community with work rights and access to health cover significantly improves psychiatric symptoms in forced refugees [21].

In contrast to previous findings [2], that reported that consequences of illness are associated positively with medication adherence, the findings of this study indicated that, the impact and seriousness of illness is inversely related to medication taking behaviour. This finding is counter-intuitive, however, the consequences may elicit an emotional response (e.g., feelings of hopelessness) and maladaptive coping, which could explain poorer adherence [50].

A majority of studies reported in the literature have examined several factors that influence medication adherence, such as age, gender, comorbidities and ethnicity [51–53]. However, none of these factors can be modified to enhance medication adherence. Patients' cognitive models of their illnesses are, by their nature, private. Patients are often reluctant to discuss beliefs about their illnesses in medical consultations because they fear being seen as misinformed [42]. However, these perceptions are amendable to counselling by health care providers and should be targeted for intervention to enhance medication adherence.

Our study also revealed suboptimal adherence levels in the refugee group, highlighting the need for urgent attention that may improve the overall quality of life for vulnerable patients who arrive in Australia. Most refugees in this study come from countries that are currently involved in war or conflict. These countries experience unique and severe complications within their health system, hospitals and healthcare professions, thus resulting in a high degree of uncertainty regarding the safety of seeking healthcare services [54].

Future interventions to improve medication adherence should address modifying illness perceptions through different approaches. Programs that close gaps in educational outcomes between ethnic minority populations, such as refugees, and majority populations are needed to promote health equity [55]. These programs may promote refugees' understanding of their health, and illnesses. Also, they may enhance refugees' ability to control their illness and overall, increase the adherence to medications. On the other hand, cultural brokering is needed to bridge the gap in cultural competent care and in workplace harmony. Actions include consultation and collaboration with a transcultural nurse generalist to help nursing and other health personnel become more culturally aware, sensitive, and develop cultural competence for culturally diverse populations, such as refugees or migrants [56].

Another approach to promote refugees' personal control perceptions, can be achieved through shared decision making, that allows them to feel understood and valued, and helps to develop a sense of independence and efficacy [57]. Understanding of the way in which cultural factors affect the incidence, course, experience and outcome of disease is crucial for clinical medicine. Religious medicine is grounded in the Arabic Middle East in the logic of healing through the power of the sacred words, the touch of holy men, or the manipulation of impurity [9]. Therefore, healthcare providers should take into account refugees and migrants' cultural background, and religious beliefs and their impact on illness perceptions, specifically their causal attributions of hypertension, to achieve better medication adherence. This highlights also, the need for tailored educational strategies about the possible factors underlying hypertension onset and how taking medication continuously can positively impact its course. Healthcare providers also, should understand the differences between refugees and migrants, and how they perceive their hypertension. Acquiring an awareness of each population's illness perceptions may help healthcare providers to identify gaps between their own understanding and the expectations of refugees and migrants about their illness and treatments.

Consequently, this may lead to the provision of more optimal health care that meets the needs and expectations of each population.

Refugees may be disadvantaged further by language barriers and lower levels of education, which may contribute to difficulties in accessing health care facilities and seeking advice only from health professionals. Migrants in Australia must meet English language requirements prior to migration [58]. Arabic speaking refugees might benefit, during medical encounters, from receiving consumer medicine information sheets in Arabic, designed specifically for those with low English-language literacy levels to augment counselling process [24].

This study has unique and specific implications for healthcare providers in general and community pharmacists specifically, since they are often the first to interact with refugees, and as experts on medication they can make a difference in the lives of these patients. Thus, it is essential to use patient-friendly educational materials to enhance refugees' understanding, and adherence to therapeutic regimen [59].

For future studies, the addition of qualitative methods, to evaluate illness perceptions is suggested. In addition evaluating the experience of newly diagnosed refugees and migrants access health services and if it relates to their cultural behaviours and taking medications may be useful. Another aspect that might be important to address in the future research, is the role of the social support in treatment adherence and the emotional factors that might be related this.

This study has some limitations. A self-reported measure was used to determine medication adherence, and illness perceptions; hence, there might be overestimation of adherence and perceptions. However, over 50% of participants reported low level of adherence, suggesting that overestimation was not a major limitation in this study. Emotional representation of illness perceptions was not measured. We chose to focus on cognitive domain of illness perception, as it remains central to medication adherence interventions.

## Conclusion

Patients' illness perceptions have a significant influence on medication adherence. Differences between refugees and migrants regarding their perceptions, and adherence should be assessed prior to providing healthcare counselling, and medical advice. This study draws attention to enhancing medication adherence amongst hypertensive refugees and migrants, by understanding and managing their negative illness perceptions. This study also gives an insight to the need for future interventional studies to promote medication adherence amongst vulnerable patients, by improving their personal, treatment control and increasing their understanding of the actual risk factors underlining hypertension.

## Acknowledgments

The authors would like to thank the following agencies: Victorian Arabic Social Services, Kangan institute, Iraqi women's social groups, and the administrators of the included Facebook groups, for assisting in recruitment of participants. We would also like to thank all the participants who freely and cheerfully gave up their time to complete the survey.

## Author Contributions

**Conceptualization:** Wejdan Shahin, Gerard A. Kennedy, Wendell Cockshaw, Ieva Stupans.

**Data curation:** Wejdan Shahin, Ieva Stupans.

**Formal analysis:** Wejdan Shahin, Gerard A. Kennedy, Wendell Cockshaw.

**Methodology:** Wejdan Shahin, Gerard A. Kennedy, Wendell Cockshaw, Ieva Stupans.

**Resources:** Wejdan Shahin.

**Software:** Wejdan Shahin.

**Supervision:** Gerard A. Kennedy, Ieva Stupans.

**Writing – original draft:** Wejdan Shahin.

**Writing – review & editing:** Gerard A. Kennedy, Wendell Cockshaw, Ieva Stupans.

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
