## [Decision Letter · Decision Letter 0]

20 Nov 2019

PONE-D-19-28766

The role of refugee and migrant migration status on medication adherence: Mediation through illness perceptions

PLOS ONE

Dear Mrs. Shahin,

Thank you for submitting your manuscript to PLOS ONE. After careful consideration, we feel that it has merit but does not fully meet PLOS ONE’s publication criteria as it currently stands. Therefore, we invite you to submit a revised version of the manuscript that addresses the points raised during the review process.

We would appreciate receiving your revised manuscript by Jan 04 2020 11:59PM. To enhance the reproducibility of your results, we recommend that if applicable you deposit your laboratory protocols in protocols.io, where a protocol can be assigned its own identifier (DOI) such that it can be cited independently in the future. For instructions see: http://journals.plos.org/plosone/s/submission-guidelines#loc-laboratory-protocols

We look forward to receiving your revised manuscript.

Kind regards,

Baltica Cabieses, PhD

Academic Editor

PLOS ONE

Journal Requirements:

Additional Editor Comments:

Dear authors

Thank you for submiting this interesting manuscript to Plos One. It tackles a relevant issue in an underserved population like refugees. It looks at a prevalent disease (Hypertension) and produces interesting findings related to access to healthcare, in particular to medication adherence in hypertension mediated by Middle Eastern refugees’ and migrants’ illness perceptions, in Australia.

The mauscrit has been revised by two independent reviewers, and they have added useful comments for improvement. Please revise them carefully, a they can add depth and quality to your study.

In addition, I would like to ask you to consider the following recommendations:

1) Methods: please discuss and justify the sample size, especially the estimated medium effect size d= 0.3.

2) Analysis: please explain how heterogeneity/variance was dealt with in the analysis.

3) Discussion: I think that the ideas that reviewer 2 is suggesting as future studies, could actually be added in the discussion section.

Thank you and I look forward to reading your revised manuscript.

Reviewers' comments:

Reviewer's Responses to Questions

**Comments to the Author**

1. Is the manuscript technically sound, and do the data support the conclusions?

Reviewer #1: Yes

Reviewer #2: Yes

2. Has the statistical analysis been performed appropriately and rigorously? 

Reviewer #1: I Don't Know

Reviewer #2: Yes

3. Have the authors made all data underlying the findings in their manuscript fully available?

Reviewer #1: Yes

Reviewer #2: Yes

4. Is the manuscript presented in an intelligible fashion and written in standard English?

Reviewer #1: Yes

Reviewer #2: Yes

5. Review Comments to the Author

Reviewer #1: This article describes an original topic: associations between illness perceptions and medication adherences in migrants and refugee population. In the first place is appreciable that research’s objective takes in count the difference among both groups, because, as the authors mention, usually they have been considered as a single population. However, the major problem of the article is the assumption that illness perception and medical adherence are individual behaviours. Even though “common sense model of illness perception” is clearly described, this model assumes that causal attributions of the illness are directly connected with how patients make sense of their symptoms. This model is based on causality, which is the origin of western rationality, and biomedical explanatory models. Different authors from medical anthropology (Arthur Kleiman, Byron Good, Evans Pritchard, Carolyn Sargent) demonstrates how causality is not the unique and the central model to explain illness in different cultures. Relationships between God, environment, spirits and other human beings are some relevant variables used to explain illness in non-western cultures as, for example, the Arabic ones. In these cultures, “fate” or “stress” can involve a cultural significance, that represents a different perception of illness. Cultural influence is not considered in the article while the authors assert the idea, not sufficiently explained, that “patients’ cognitive models of their illnesses are, by their nature, private” (475-476). In order to understand the meanings of illness such as hypertension in Arabic population, its highly recommended to include the work of Byron Good based on Iranian perception of hearth disease called: "The Heart of What's the Matter: The Structure of Medical Discourse in a Provincial Iranian Town” or Good, B. J., & Good, M. J. D. (1982). “Toward a meaning-centered analysis of popular illness categories: “fright illness” and “heart distress” in Iran”. In second place it’s necessary to pay attention to the differences perceived between refugees and migrants, specially regarding “individual control” of their illness. The experience of war, violence, and forced migration has enormous consequences on the sense of control of the lives of refugees (1995. Desjarlais, Robert, Leon Eisenberg, Byron J. Good, and Arthur Kleinman. World Mental Health: Problems and Priorities in Low Income Countries. New York: Oxford University Press; 2015. Devon Hinton and Byron Good, eds. Culture and PTSD. Philadelphia: University of Pennsylvania Press.). This experience of total loss of control, has evidently consequences on their conceptions of life and death, illness, fate, and how they perceive their bodies. Nothing of that is mentioned in the discussion, reproducing the idea that medical adherence is result uniquely of individual behaviour. I recommend including a reflection about it. Finally, I suggest evaluating some statements as 476-478, considering the use of cultural brokers in health care settings and their outcomes in migrants and refugees’ capacity to communicate in the clinical setting (for example: Jeffreys, M. R. (2005). Clinical nurse specialists as cultural brokers, change agents, and partners in meeting the needs of culturally diverse populations. Journal of Multicultural Nursing & Health, 11(2), 41).

Reviewer #2: It is a very interesting research.

For future studies:

-It would be interesting to evaluate the perceptions with mix methodology and add qualitative methodology to the next version of this research.

- Know the participants experiences about the access to health services when they are first diagnosed (if it is cultural sensitive for example) and if it is related to the adherence of the treatment.

-Also It would be important to evaluate the social support in the treatment and the emotional factors that could be related.

6. PLOS authors have the option to publish the peer review history of their article (what does this mean?). If published, this will include your full peer review and any attached files.

Reviewer #1: No

Reviewer #2: Yes: Daniela Pacheco Olmedo

---

## [Author Response · Author response to Decision Letter 0]

11 Dec 2019

Thank you for considering this paper. Below are the comments and the responses to both of the reviewers and the editor. 

Please provide additional details regarding participant consent. In the ethics statement in the Methods and online submission information, please ensure that you have specified (1) whether consent was informed and (2) what type you obtained (for instance, written or verbal, and if verbal, how it was documented and witnessed). If your study included minors, state whether you obtained consent from parents or guardians. If the need for consent was waived by the ethics committee, please include this information. Participant information has been provided (attached) and then implied consent has been given through moving onto and returning the anonymous survey.

-We have added this clearly in the method section. 

We note that you have stated that you will provide repository information for your data at acceptance. Should your manuscript be accepted for publication, we will hold it until you provide the relevant accession numbers or DOIs necessary to access your data. If you wish to make changes to your Data Availability statement, please describe these changes in your cover letter and we will update your Data Availability statement to reflect the information you provide. 

-We will provide the DOI or the accession number once we are informed that this paper has been accepted. 

Methods: please discuss and justify the sample size, especially the estimated medium effect size d= 0.3. 

-This has been corrected and justified, by referring to Cohen’s D statistics.

Analysis: please explain how heterogeneity/variance was dealt with in the analysis. 

-Heterogeneity of variance was tested using Levenes test. This has been mentioned in the analysis section.

Discussion: I think that the ideas that reviewer 2 is suggesting as future studies, could actually be added in the discussion section. 

-Done in the discussion section. 

The major problem of the article is the assumption that illness perception and medical adherence are individual behaviours. Even though “common sense model of illness perception” is clearly described, this model assumes that causal attributions of the illness are directly connected with how patients make sense of their symptoms. This model is based on causality, which is the origin of western rationality, and biomedical explanatory models.

Different authors from medical anthropology (Arthur Kleiman, Byron Good, Evans Pritchard, Carolyn Sargent) demonstrates how causality is not the unique and the central model to explain illness in different cultures. 

-It is very clear that causality is a universal form of thinking and not just Western. What is attributed as the cause can be influenced by religion. We think the causality is a central model to explain illness, but the cause attributed is often a correlation rather than a true cause. 

Relationships between God, environment, spirits and other human beings are some relevant variables used to explain illness in non-western cultures as, for example, the Arabic ones. In these cultures, “fate” or “stress” can involve a cultural significance, that represents a different perception of illness. Cultural influence is not considered in the article while the authors assert the idea, not sufficiently explained, that “patients’ cognitive models of their illnesses are, by their nature, private” (475-476). In order to understand the meanings of illness such as hypertension in Arabic population, its highly recommended to include the work of Byron Good based on Iranian perception of heart disease called: "The Heart of What's the Matter: The Structure of Medical Discourse in a Provincial Iranian Town” or Good, B. J., & Good, M. J. D. (1982). “Toward a meaning-centered analysis of popular illness categories: “fright illness” and “heart distress” in Iran” 

-The significance of culture has been mentioned in the introduction and also discussed in the discussion section, by using one of the references that you mentioned. 

Introduction: 

“A number of studies have also assumed the view that disease may be a response to social stresses and/or life events and is shaped in part by the nature of the cultural label which is applied to a person's condition”

Discussion:

Understanding of the way in which cultural factors affect the incidence, course, experience and outcome of disease is crucial for clinical medicine. Religious medicine is grounded in the Arab Middle East via the logic of healing through the power of the sacred words, the touch of holy men, or the manipulation of impurity. 

In second place it’s necessary to pay attention to the differences perceived between refugees and migrants, specially regarding “individual control” of their illness. The experience of war, violence, and forced migration has enormous consequences on the sense of control of the lives of refugees (1995. Desjarlais, Robert, Leon Eisenberg, Byron J. Good, and Arthur Kleinman. World Mental Health: Problems and Priorities in Low Income Countries. New York: Oxford University Press; 2015. Devon Hinton and Byron Good, eds. Culture and PTSD. Philadelphia: University of Pennsylvania Press.). 

-This statement has been included to highlight the differences regarding individual control. “Refugees have been forced into a situation where responsibility for and control over their own lives has been taken away from them. Their existence and future is uncertain, and many experience a constant fear of being deported. The powerlessness they experienced generates uncertainty that has negative implications for health”. 

Unemployment, and denial of access to health services are risk factors for psychiatric morbidity, and chronicity of health conditions. Refugees constitute a particularly high risk group.

This experience of total loss of control, has evidently consequences on their conceptions of life and death, illness, fate, and how they perceive their bodies. Nothing of that is mentioned in the discussion, reproducing the idea that medical adherence is result uniquely of individual behaviour. I recommend including a reflection about it. 

-Four references have been used to cite evidence about the consequences of loss of control in the discussion part. “More positive beliefs about the sense of internal control have been associated with coping strategies which are generally considered more adaptive such as positive reinterpretation, seeking social support and actively trying to tackle the problems. Traumatized refugees who experienced war, forced migration or violence perceive absence of control over their lives — this can contribute to poor health as diet, exercise and medical treatment are neglected.

Social support plays an important role in determining treatment uptake, recovery and adherence. Refugees who have been taken away from their friends and families, lack social support thus, worse adherence to medication and health recovery would be expected. In literature, reduced posttraumatic stress was associated with securing work rights and health cover. Living in the community with work rights and access to health cover significantly improves psychiatric symptoms in forced refugees. 

Finally, I suggest evaluating some statements as 476-478, considering the use of cultural brokers in health care settings and their outcomes in migrants and refugees’ capacity to communicate in the clinical setting (for example: Jeffreys, M. R. (2005). Clinical nurse specialists as cultural brokers, change agents, and partners in meeting the needs of culturally diverse populations. Journal of Multicultural Nursing & Health, 11(2), 41). 

-The role of cultural brokers has been discussed in the discussion section, highlighting the importance of having cultural brokers to bridge the gap in cultural competent care. 

For future studies:

It would be interesting to evaluate the perceptions with mix methodology and add qualitative methodology to the next version of this research.

Know the participants experiences about the access to health services when they are first diagnosed (if it is cultural sensitive for example) and if it is related to the adherence of the treatment.

Also It would be important to evaluate the social support in the treatment and the emotional factors that could be related. 

-This has been included before the last paragraph of the discussion: “For future studies, the addition of qualitative methods, to evaluate illness perceptions is suggested. In addition evaluating the experience of newly diagnosed refugees and migrants access health services and if it relates to their cultural behaviours and taking medications may be useful. Another aspects that might be important to address in the future research, is the role of the social support in treatment adherence and the emotional factors that might be related this”.

---

## [Editor Report · Decision Letter 1]

18 Dec 2019

The role of refugee and migrant migration status on medication adherence: Mediation through illness perceptions

PONE-D-19-28766R1

Dear Dr. Shahin,

We are pleased to inform you that your manuscript has been judged scientifically suitable for publication and will be formally accepted for publication once it complies with all outstanding technical requirements.

With kind regards,

Baltica Cabieses, PhD

Academic Editor

PLOS ONE

Additional Editor Comments (optional):

The revision has adequately addressed the reviewers and editor´s comments. 
---

## [Editor Report · Acceptance letter]

26 Dec 2019

PONE-D-19-28766R1 

The role of refugee and migrant migration status on medication adherence: Mediation through illness perceptions 

Dear Dr. Shahin:

I am pleased to inform you that your manuscript has been deemed suitable for publication in PLOS ONE. Congratulations! Your manuscript is now with our production department. 

With kind regards,

on behalf of

Dr. Baltica Cabieses 

Academic Editor

PLOS ONE